# Convergence of adaptive algorithms for constrained weakly convex optimization

**Ahmet Alacaoglu**
University of Wisconsin-Madison
alacaoglu@wisc.edu

**Yura Malitsky**
Linköping University
yurii.malitskyi@liu.se

**Volkan Cevher**
EPFL
volkan.cevher@epfl.ch

## Abstract

We analyze the adaptive first order algorithm AMSGrad, for solving a constrained stochastic optimization problem with a weakly convex objective. We prove the $\tilde{\mathcal{O}}(t^{-1/2})$ rate of convergence for the squared norm of the gradient of Moreau envelope, which is the standard stationarity measure for this class of problems. It matches the known rates that adaptive algorithms enjoy for the specific case of unconstrained smooth nonconvex stochastic optimization. Our analysis works with mini-batch size of 1, constant first and second order moment parameters, and possibly unbounded optimization domains. Finally, we illustrate the applications and extensions of our results to specific problems and algorithms.

## 1   Introduction

Adaptive first order methods have become a mainstay of neural network training in recent years. Most of these methods build on the AdaGrad framework [14], which is a modification of online gradient descent by incorporating the sum of the squared gradients in the step size rule. Based on the practical shortcomings of AdaGrad for training neural networks, RMSprop [32] and Adam [21] proposed to use exponential moving averages for gradients and squared gradients (also known as *moment estimations*) with parameters $\beta_1$ and $\beta_2$, respectively. These methods have seen a huge practical success.

The recent work [28] identified a technical issue that affects Adam and RMSprop and proposed a new Adam-variant called AMSGrad that does not suffer from the same problem. Theoretical properties of AMSGrad, AdaGrad and their variants for nonconvex optimization problems are studied in a number of recent papers [3, 4, 11, 24, 34, 37]. These works focus on unconstrained smooth stochastic optimization, where the standard analysis framework of the stochastic gradient descent (SGD) [18] can be used. Convergence of adaptive methods for the more general setting of constrained and/or nonsmooth stochastic nonconvex optimization has remained open, while these settings have broad practical applications [7, 13, 20, 25, 27, 33].

In this work, we take a step towards this direction and establish the convergence of AMSGrad for solving the problem

$$\min_{x \in \mathcal{X}} \left\{ f(x) = \mathbb{E}_{\xi \sim \mathsf{P}} \left[ f(x; \xi) \right] \right\}, \tag{1}$$

where $f \colon \mathbb{R}^d \to \mathbb{R}$ is $\rho$-weakly convex, $\mathcal{X} \subseteq \mathbb{R}^d$ is closed convex, and $\xi$ is a r.v. following a fixed unknown distribution $\mathsf{P}$. This template captures the setting of previous analyses when $f$ is $L$-smooth, as this implies $L$-weak convexity, and $\mathcal{X} = \mathbb{R}^d$. However, there exist many applications when $\mathcal{X} \neq \mathbb{R}^d$ [20, 25, 27, 33] or when $f$ is not $L$-smooth [7, Section 2.1],[13, 15].

Constrained stochastic minimization with nonconvexity presents challenges not met in the convex setting [5, 19]. In particular, until the recent work [7], even for SGD, increasing mini-batch sizes were required for convergence in constrained nonconvex optimization. To study the behavior of AMSGrad for solving (1), we build on the analysis framework of [7].

35th Conference on Neural Information Processing Systems (NeurIPS 2021).

| | $f$ | Constraints | $\beta_1$ | mini-batch size | Diagonal | Adaptive |
|---|---|---|---|---|---|---|
| [4] | $L$-smooth | $\times$ | const. | 1 | ✓ | ✓ |
| [5] | $L$-smooth | ✓ | 0 | $\sim \sqrt{t}$ | ✓ | ✓ |
| [7] | $\rho$-weak. cvx. | ✓ | 0 | 1 | $\times$ | $\times$ |
| [26] | $\rho$-weak. cvx. | ✓ | const. | 1 | $\times$ | $\times$ |
| This work | $\rho$-weak. cvx. | ✓ | const. | 1 | ✓ | ✓ |

Table 1: Comparison with adaptive methods for smooth nonconvex optimization and SGD-based methods for weakly convex optimization. Column "diagonal" refers to coordinatewise step sizes and "adaptive" refers to step sizes depending on observed gradients á la AdaGrad.

**Contributions.** We show that AMSGrad achieves $\mathcal{O}(\log(T)/\sqrt{T})$ rate for near-stationarity, see (5), for solving (1). Key specifications for this result are the following:

- We can use a mini-batch size of 1.
- We can use constant moment parameters $\beta_1, \beta_2$ which are used in practice [1, 4, 21, 28].
- We do not assume boundedness of the domain $\mathcal{X}$.

Next, we particularize our results for constrained optimization with $L$-smooth objectives and for a variant of RMSprop. We also extend our analysis for the scalar version of AdaGrad with first order moment estimation. For easy reference, we compare our results with state-of-the-art in Table 1. Finally, in a numerical experiment for robust phase retrieval, we observe that AMSGrad is more robust to variation of initial step sizes, compared to SGD and SGD with momentum.

---

**Algorithm 1** AMSGrad [28]

---

**Input:** $x_1 \in \mathcal{X}$, $\alpha_t = \frac{\alpha}{\sqrt{t}}$, for $t \geq 1$, $\alpha > 0$,
$\beta_1 < 1, \beta_2 < 1$,
$m_0 = v_0 = 0, \hat{v}_0 = \delta\mathbb{1}, 1 \geq \delta > 0$.
**for** $t = 1, 2 \ldots T$ **do**
    Sample $\xi_t \sim \mathsf{P}$ iid and set $g_t$ such that $\mathbb{E}_{\xi_t}[g_t] \in \partial f(x_t)$ (see Assumption 1)
    $m_t = \beta_1 m_{t-1} + (1 - \beta_1)g_t$
    $v_t = \beta_2 v_{t-1} + (1 - \beta_2)g_t^2$
    $\hat{v}_t = \max(\hat{v}_{t-1}, v_t)$
    $x_{t+1} = P_{\mathcal{X}}^{\hat{v}_t^{1/2}}(x_t - \alpha_t \hat{v}_t^{-1/2} m_t)$
**end for**
**Output:** $x_{t^*}$, where $t^*$ is selected uniformly at random from $\{1, \ldots, T\}$.

---

## 1.1 Examples of weakly convex problems

The class of problems we consider in this paper include constrained problems with $L$-smooth objectives which are, for example, studied in [5] in the context of adversarial attacks and adaptive methods. Other important examples with weak convexity are composite objectives $h(c(x))$, where $h$ is a convex Lipschitz continuous function and $c$ is a smooth map with Lipschitz continuous Jacobian. Concrete examples of weakly convex problems are listed in [7, Section 2.1], which include robust phase retrieval, sparse dictionary learning, Conditional Value-at-Risk, and many others.

## 1.2 Related work

Adaptive algorithms based on AdaGrad [14] and Adam [1, 21, 28] are classically analyzed in online optimization framework with convex objective functions. Recent works studied the behavior of these methods for nonconvex optimization [2–4, 11, 24, 34, 36, 37]. The common characteristic of these results is that they are based on the well established proof templates of SGD [18] that only works in the simplest case of unconstrained smooth stochastic minimization. Moreover, as mentioned in [1], unconstrained optimization makes it easier to use a constant $\beta_1$ parameter in Adam-type methods. In particular, many results for constrained optimization require a fast diminishing schedule for $\beta_1$ parameter, while a constant parameter is used in practice [5, 21, 28].

The specific case of (1) with $L$-smooth $f$ is studied by [5], where the authors proposed a zeroth order variant of AMSGrad. This result applies for the specific case of $\beta_1 = 0$ which corresponds to a variant of RMSprop [28, 32]. More importantly, since its analysis follows the one of [19], increasing mini-batch sizes of the order $\sqrt{t}$ are required [5, Theorem 2].

As also mentioned in [5, 7], analysis of SGD for constrained problems introduces specific difficulties that are not observed in the convex case. Due to this, classical works analyzing SGD for nonconvex constrained optimization used large mini-batches to ensure convergence [19]. Showing convergence for SGD for constrained optimization with a single sample had been an open question until [7] gave a positive answer in the framework of weakly convex stochastic optimization, which includes constrained smooth stochastic optimization as a special case.

Weakly convex optimization is well studied with SGD based methods [7, 10, 15]. Recent work by [26], considers momentum SGD for solving (1). However, this algorithm *(i)* does not use momentum with $\beta_2$ and *(ii)* uses non-adaptive, scalar, fixed step size: in the notation of Algorithm 1, $\hat{v}_t = 1$, $\alpha_t = \alpha/\sqrt{T}$. These make the algorithm less practical, while simpler for analysis. Moreover the original work of Davis and Drusvyatskiy [7] analyzed standard SGD with no momentum or adaptive step sizes. As incorporating these mechanisms introduce more error terms, we need tighter estimations (for example see (9)) and techniques to handle time-dependent adaptive and diagonal step sizes.

The difficulty of handling the combination of nonconvexity, adaptive step sizes, momentum and constrained sets is mentioned in [5, Section 4.3]. In particular, in terms of our analysis, *(i)* adaptive step size introduces coupling between the step sizes, iterates, and the *proximal point* defined in (4); *(ii)* time-dependent diagonal step size requires an analysis framework based on variable metrics. Both of these issues were not the case in earlier works for weakly convex optimization [7, 26], and are among the key aspects of our analysis. For details, please see Lemma 1, Lemma 2 and discussions therein.

Another promising direction of research concerns nonsmooth nonconvex problems under more general assumptions. For instance, *tameness* and *Hadamard semi-differentiability* are used in [8] and [35], respectively, where convergence guarantees are established for SGD-based methods. Because of the generality of the problem class in these works, the algorithms studied there are simpler than the Adam-type algorithms considered in this paper, and the stationarity measures are less standard [35].

### 1.3 Notation

We adopt the convention of using the standard operations $ab$, $a^2$, $a/b$, $a^{1/2}$, $1/a$, $\max\{a,b\}$ as element-wise, given two vectors $a, b \in \mathbb{R}^d$. To denote $i^{\text{th}}$ element of the vector $a_t \in \mathbb{R}^d$, we use the notation $a_{t,i}$. All-ones vector is denoted as $\mathbb{1}$. Given a vector $a \in \mathbb{R}^d$, we define the matrix $\text{diag}(a) \in \mathbb{R}^{d \times d}$ as the diagonal matrix with elements of $a$ in the diagonal. For any set $\mathcal{X}$, indicator function $I_{\mathcal{X}}$ is given by $I_{\mathcal{X}}(x) = 0$ if $x \in \mathcal{X}$ and $I_{\mathcal{X}}(x) = +\infty$ otherwise.

Given the elements $v_i > 0$, $i = 1, \ldots, d$, we define a weighted norm $\|x\|_v^2 := \langle x, \text{diag}(v)x \rangle$. The weighted projection operator onto $\mathcal{X}$ is defined as $P_{\mathcal{X}}^v(x) = \text{argmin}_{y \in \mathcal{X}} \|y - x\|_v^2$. A standard property of this operator is nonexpansiveness: $\forall x, y \in \mathbb{R}^d$, $\|P_{\mathcal{X}}^v(y) - P_{\mathcal{X}}^v(x)\|_v \leq \|y - x\|_v$.

Due to nonconvexity, we cannot use standard definition of subgradients to form a global under-estimator. *Regular subdifferential*, denoted as $\partial f$, for nonconvex functions [30, Ch. 8] is defined as the set of vectors $q \in \mathbb{R}^d$ such that, $\forall x, y \in \mathbb{R}^d$, $q \in \partial f(x)$ if

$$f(y) \geq f(x) + \langle y - x, q \rangle + o(\|y - x\|), \quad \text{as } y \to x. \tag{2}$$

When $f$ is convex, this reduces to standard definition of a subdifferential and when $f$ is differentiable, this set coincides with $\{\nabla f(x)\}$. We say that $f$ is *$\rho$-weakly convex* w.r.t. $\|\cdot\|$, if $f(x) + \frac{\rho}{2}\|x\|^2$ is convex. An equivalent representation for weakly convex functions is that, $\forall x, y \in \mathbb{R}^d$, where $q \in \partial f(x)$ [7, Lemma 2.1],

$$f(y) \geq f(x) + \langle y - x, q \rangle - \frac{\rho}{2}\|y - x\|^2. \tag{3}$$

Moreover, we say $f$ is *$L$-smooth*, if it holds that, $\forall x, y \in \mathbb{R}^d$, $\|\nabla f(x) - \nabla f(y)\| \leq L\|x - y\|$. Given random iterates $x_1, \ldots, x_t$, we denote the filtration generated by these realizations as $\mathcal{F}_t = \sigma(x_1, \ldots, x_t)$, and the corresponding conditional expectation as $\mathbb{E}_t[\cdot] = \mathbb{E}[\cdot|\mathcal{F}_t]$. By the law of total expectation, $\mathbb{E}[\mathbb{E}_t[\cdot]] = \mathbb{E}[\cdot]$. We also sometimes use $\mathbb{E}_\xi$ to denote expectation w.r.t. randomness of $\xi$.

We now present the assumptions of our analysis.

---

**Assumption 1.**
- $f\colon \mathbb{R}^d \to \mathbb{R}$ *is $\rho$-weakly convex with respect to norm $\|\cdot\|$.*
- *The set $\mathcal{X} \subseteq \mathbb{R}^d$ is convex and closed.*
- *We can obtain iid samples $\xi_t \sim \mathsf{P}$, and $g_t$ such that $\mathbb{E}_{\xi_t}[g_t] \in \partial f(x_t)$ and $\|g_t\|_\infty \leq G, \forall t$.*
- *$f$ is lower bounded: $f^\star \leq f(x), \forall x \in \mathcal{X}$.*

---

A few remarks are in order for Assumption 1. First, we do not require boundedness of the domain $\mathcal{X}$. Second, weak convexity assumption is weaker than smoothness assumption on $f$ and the assumption of bounded gradients is standard for adaptive algorithms [3, 4, 11]. In principle, it is possible to relax the bounded stochastic subgradient assumption to the weaker requirement $\mathbb{E}\|g_t\|^2 \leq G$ as in [37, Remark 6. (ii)] with a slightly worse and complicated convergence rate. For simplicity, we use Assumption 1. The third item in Assumption 1 concerning sampling a stochastic oracle is standard in the weakly convex optimization literature [7, Assumption A2], [26, Assumption A1].

**Remark 1.** *We note that when $f$ is $\rho$-weakly convex w.r.t. $\|\cdot\|$, then it is $\frac{\rho}{\sqrt{\delta}}$-weakly convex w.r.t. $\|\cdot\|_{\hat{v}_t^{1/2}}, \forall t$, since $\hat{v}_{t,i} \geq \delta > 0$ (see Algorithm 1). We denote $\hat{\rho} = \frac{\rho}{\sqrt{\delta}}$.*

It is easy to verify this remark by noticing that $x \mapsto f(x) + \frac{\rho}{2}\|x\|^2$ is convex and $\frac{\hat{\rho}}{2}\|x\|_{\hat{v}_t^{1/2}}^2 \geq \frac{\rho}{2}\|x\|^2$.

## 2 Algorithm

We analyze the algorithm AMSGrad (see Algorithm 1) proposed in [28]. On top of Adam [21], it includes a step to ensure monotonicity of the exponential moving average of squared gradients. It is standard in stochastic nonconvex optimization to output a randomly selected iterate [7, 18, 19], which we also adopt. We next define the composite objective

$$\varphi(x) = f(x) + I_{\mathcal{X}}(x).$$

For nonsmooth problems, the standard stationarity measures such as the norm of (sub)gradients are no longer applicable, see [7, 26] and [13, Section 4]. This motivates the following definitions that, as we show below, relate to a relaxed form of stationarity. Based on $\varphi$ and a parameter $\bar{\rho} > 0$, we define the proximal point of $x_t$ and the Moreau envelope, respectively as

$$\hat{x}_t = \text{prox}_{\varphi/\bar{\rho}}^{\hat{v}_t^{1/2}}(x_t) = \underset{y}{\operatorname{argmin}} \left\{ \varphi(y) + \frac{\bar{\rho}}{2}\|y - x_t\|_{\hat{v}_t^{1/2}}^2 \right\}, \tag{4}$$

$$\varphi_{1/\bar{\rho}}^t(x_t) = \min_y \left\{ \varphi(y) + \frac{\bar{\rho}}{2}\|y - x_t\|_{\hat{v}_t^{1/2}}^2 \right\}.$$

We compare the definitions with that of [7]. Due to the use of variable metric $\hat{v}_t$ in adaptive methods, we have a time dependent Moreau envelope, where the corresponding vector $\hat{v}_t$ is used for defining the norm. Important considerations for these quantities are the uniqueness of $\hat{x}_t$ and the smoothness of $\varphi_{1/\bar{\rho}}^t$. As we shall see now, choice of $\bar{\rho}$ is critical for ensuring these. In light of Remark 1, selecting $\bar{\rho} > \hat{\rho} = \frac{\rho}{\sqrt{\delta}}$, and by using the arguments in [7, for example, Lemma 2.2], [29, Theorem 31.5], it follows $\hat{x}_t$ is unique and $\varphi_{1/\bar{\rho}}^t$ is smooth with the gradient $\nabla\varphi_{1/\bar{\rho}}^t(x_t) = \bar{\rho}\hat{v}_t^{1/2}(x_t - \hat{x}_t)$.

**Near stationarity.** Near-stationarity conditions follow from the optimality condition of $\hat{x}_t$: $0 \in \partial\varphi(\hat{x}_t) + \bar{\rho}\hat{v}_t^{1/2}(\hat{x}_t - x_t)$, where we also use $\hat{v}_{t,i} \leq G^2$:

$$\begin{cases} \|x_t - \hat{x}_t\|_{\hat{v}_t^{1/2}}^2 = \frac{1}{\bar{\rho}^2}\|\nabla\varphi_{1/\bar{\rho}}^t(x_t)\|_{\hat{v}_t^{-1/2}}^2 \\ \text{dist}^2(0, \partial\varphi(\hat{x}_t)) \leq G\|\nabla\varphi_{1/\bar{\rho}}^t(x_t)\|_{\hat{v}_t^{-1/2}}^2 \\ \varphi(\hat{x}_t) \leq \varphi(x_t). \end{cases} \tag{5}$$

Consistent with previous literature for weakly convex optimization [7, 26], we state the convergence guarantees in terms of the norm of the gradient of Moreau envelope. Given (5), this means that the iterate $x_t$ is near stationary: it is close to its proximal point $\hat{x}_t$ and $\hat{x}_t$ is approximately stationary.

## 3 Convergence

### 3.1 Main result

We start with our main theorem that shows that the norm of the gradient of Moreau envelope converges to 0 at the claimed rate, resulting in near-stationarity of $x_{t^*}$ as in (5), see Remark 1 for definition of $\hat{\rho}$.

**Theorem 1.** *Let Assumption 1 hold. Let $\beta_1 < 1$, $\beta_2 < 1$, $\gamma = \frac{\beta_1^2}{\beta_2} < 1$, $\bar{\rho} = 2\hat{\rho}$. Then, for iterate $x_{t^*}$ generated by Algorithm 1, it follows that*

$$\mathbb{E}\|\nabla\varphi_{1/\bar{\rho}}^{t^*}(x_{t^*})\|_{\hat{v}_{t^*}^{-1/2}}^2 \leq \frac{2dG}{\alpha\sqrt{\delta T}(1-\beta_1)}\left[C_1 + (1+\log T)C_2 + C_3\right],$$

*where $C_1 = 8\rho\alpha G + \frac{1}{dG}\left(\varphi_{1/\bar{\rho}}^1(x_1) - f^\star\right)$, $C_2 = \frac{2\rho\alpha^2}{\sqrt{(1-\beta_2)(1-\gamma)}}\left(\frac{10G}{\sqrt{\delta}}\right)$, $C_3 = \frac{8G}{\rho}\sum_{i=1}^d \mathbb{E}\hat{v}_{T+1,i}^{1/2}$.*

The bound in Theorem 1 has complicated constants as it is usual for adaptive algorithms in nonconvex case [3, 4]. These constants are slightly simplified and the proof of Theorem 1 in Appendix A includes the non-simplified version. Next, we explain and interpret the bound in terms of dependence to key parameters.

### 3.2 Discussion on Theorem 1

In the context of near-stationarity (5), Theorem 1 states that to have $x_{t^*}$ in Algorithm 1 such that $\|\nabla\varphi_{1/\bar{\rho}}^{t^*}(x_{t^*})\|_{\hat{v}_t^{-1/2}} \leq \epsilon$, we require $\tilde{\mathcal{O}}(\epsilon^{-4})$ iterations. This matches the known complexities for adaptive methods in unconstrained smooth stochastic optimization [1, 3, 4, 11, 24, 34, 36, 37], and SGD-type methods in weakly convex optimization [7, 26].

Our first remark is about the metric of the norm used for the gradient of Moreau envelope in Theorem 1. We then discuss the dependence of our bound w.r.t. important quantities.

**Remark 2.** *By (5), one has $\|\nabla\varphi_{1/\bar{\rho}}^{t^*}(x_{t^*})\|_{\hat{v}_{t^*}^{-1/2}}^2 = \bar{\rho}^2\|x_{t^*}-\hat{x}_{t^*}\|_{\hat{v}_{t^*}^{1/2}}^2$. We note that $\|x_{t^*}-\hat{x}_{t^*}\|_{\hat{v}_{t^*}^{1/2}}^2 \geq \sqrt{\delta}\|x_{t^*}-\hat{x}_{t^*}\|^2$ as $\hat{v}_{t,i} \geq \delta$. It also holds that $\hat{v}_{t,i} \leq G^2$. Therefore, one can convert our guarantees to $\|x_{t^*}-\hat{x}_{t^*}\|^2$ or $\|\nabla\varphi^{t^*}(x_{t^*})\|$ by multiplying the right hand side by appropriate quantities depending on $\delta$ or $G$. We leave the result with the metric, as $\delta$ and $G$ are the worst case bounds.*

**Knowledge of $\rho$.** To run the algorithm, one does not need to know the weak convexity parameter $\rho$. The parameters $\bar{\rho}$ and $\hat{\rho}$ are merely for analysis purposes [8, 26], and the convergence rate holds for any choice of step size $\alpha_t$, independent of $\rho$.

**Dependence w.r.t. $\beta_1$.** Comparing with the previous work, the scaling of our bound in terms of $\beta_1$ is $(1-\beta_1)^{-1}$ matching the dependence for the unconstrained setting [1, 11].

**Dependence w.r.t. $d$.** Standard dependence in the convergence rates of Adam-type algorithms for unconstrained case is $d/\sqrt{T}$ [1, 11].[1]

Even though in Theorem 1, the constant $C_3$ has worst case dependence $d^2$, this is merely due to assumptions. The main reason is that we do not assume boundedness of the sequence $x_t$, instead we prove the necessary result for the analysis in Lemma 1. However, this result gives a bound for $\|x_t - \hat{x}_t\|$, which is naturally dimension dependent. We used this bound in (10), where we need to bound $\|x_t - \hat{x}_t\|_\infty$.

In particular, if we had assumed a bound for $\|x_t - \hat{x}_t\|_\infty$, then in (10) we could have used it instead of Lemma 1 to have standard $d/\sqrt{T}$ in $C_3$. Boundedness assumption also would remove a factor of $\frac{1}{\sqrt{\delta}}$ in the bound, as those terms appear in the steps where we avoid boundedness assumption. However, for generality, we do not assume boundedness.

**Dependence w.r.t. $\delta$.** Our bound has a polynomial dependence of $1/\delta$ similar to [1, 3, 4]. In [11], a more refined technique from [34] is used to have a logarithmic dependence of $1/\delta$. This technique, used on the case of smooth unconstrained problems in these works, did not seem to apply to our setting.

---

[1]We note that in [3] better dependence is obtained by using step sizes in the order of $\frac{1}{\sqrt{d}}$, which we do not consider.

### 3.3 Analysis

In this section, we will flesh out the main ideas of our proof with four lemmas. We will continue with a proof sketch to show how the pieces come together.

#### 3.3.1 Preliminary results

We start with a result showing that under Assumption 1, the quantity $\|x_t - \hat{x}_t\|$ from (5) stays bounded. We remark that third term on RHS in (9) arises as a spurious term due to time-dependent diagonal step sizes, which was not the case in previous works on weakly convex optimization with scalar step sizes [26]. Our result in the next lemma is the main tool for us to avoid assuming boundedness of $\mathcal{X}$. The proof of this lemma given in Appendix A combines the definition of $\hat{x}_t$ with weak convexity.

**Lemma 1.** *Let Assumption 1 hold. Let $\bar{\rho} > \hat{\rho}$, and $\hat{v}_t \geq \delta > 0$ (see Algorithm 1). It follows that*

$$\|x_t - \hat{x}_t\|^2 \leq \hat{D}^2 := \frac{4dG^2}{\delta(\bar{\rho} - \hat{\rho})^2}.$$

A key aspect in the analysis of adaptive algorithms is the dependence of $\hat{v}_t$ and $g_t$ that couples $\hat{x}_t$ and $g_t$ (see (4)), preventing taking expectation of $\langle x_t - \hat{x}_t, g_t \rangle$ that we use to obtain the stationarity measure in the proof. As this was not the case in prior works [7, 26], we need a more refined analysis.

**Lemma 2.** *Let Assumption 1 hold. Let $q_t = \mathbb{E}_t[g_t] \in \partial f(x_t)$, then it follows that*

$$\alpha_t \mathbb{E}_t \langle x_t - \hat{x}_t, g_t \rangle \geq \alpha_t(\bar{\rho} - \hat{\rho})\mathbb{E}_t \|x_t - \hat{x}_t\|^2_{\hat{v}_t^{1/2}} - (\alpha_{t-1} - \alpha_t)\sqrt{d}\hat{D}G - \frac{\bar{\rho} - \hat{\rho}}{4\bar{\rho}}\mathbb{E}_t \|\hat{x}_t - \hat{x}_{t-1}\|^2_{\hat{v}_{t-1}^{1/2}}$$

$$- \frac{\alpha_{t-1}}{2}\mathbb{E}_t \|m_{t-1}\|^2_{\hat{v}_{t-1}^{-1/2}} - \left(\frac{1}{2} + \frac{\bar{\rho}}{\bar{\rho} - \hat{\rho}}\right)\frac{\alpha_{t-1}^2}{\sqrt{\delta}}\mathbb{E}_t \|g_t\|^2.$$

**Interpreting Lemma 2.** We review the terms in this bound to gain some intuition. The first term in the RHS is the stationarity measure (see (5)), second term will sum to a constant, fourth and fifth terms will sum to $\log(T)$ by Lemma 4. Handling the third term in RHS is not as obvious, but we can show that we can cancel it using the contribution from another part of the analysis that we detail in the full proof (see (9)).

One critical issue for Adam-type algorithms is to obtain results with constant $\beta_1$ parameter. A recent work [1] studied this problem for constrained convex problems. The following lemma also plays an important role in our analysis.

**Lemma 3.** *[1, Lemma 1] Let $m_t = \beta_1 m_{t-1} + (1 - \beta_1)g_t$. Then for any vectors $A_{t-1}$, $A_t$,*

$$\langle A_t, g_t \rangle = \frac{1}{1 - \beta_1}\left(\langle A_t, m_t \rangle - \langle A_{t-1}, m_{t-1} \rangle\right) + \langle A_{t-1}, m_{t-1} \rangle + \frac{\beta_1}{1 - \beta_1}\langle A_{t-1} - A_t, m_{t-1} \rangle.$$

This lemma derives a decomposition for handling constant $\beta_1$ parameter in the beginning of the analysis. As explained in Section 3.1 of the abovementioned paper, using a decomposition for $m_t$ later in the analysis results in a requirement of decreasing $\beta_1$, especially for constrained problems, which we would like to avoid.

Next lemma is a standard estimation used for the analysis of Adam-based methods, dating back to [21]. For easy reference we point out to [1] where this bound is included as a separate lemma with tighter estimations than previous works, due to using a constant $\beta_1$. It bounds the sum of the norms of first moment vectors multiplied by the step size sequence.

**Lemma 4.** *Let $\beta_1 < 1$, $\beta_2 < 1$, $\gamma = \frac{\beta_1^2}{\beta_2} < 1$, then it holds that*

$$\sum_{t=1}^{T} \alpha_t^2 \|m_t\|^2_{\hat{v}_t^{-1/2}} \leq \frac{(1 - \beta_1)\alpha^2}{\sqrt{(1 - \beta_2)(1 - \gamma)}}dG(1 + \log T).$$

#### 3.3.2 Proof sketch of Theorem 1

The proof sketch of the theorem is a careful combination of the preliminary results mentioned in the previous section. The sketch includes the necessary bounds, but omits the tedious estimations required in some steps. The full proof with the details is given in Appendix A.

*Proof sketch of Theorem 1.* We sum the result of Lemma 3 and use $A_1 = A_0$. with $m_0 = 0$. We note that we have $A_t = \bar{\rho}\alpha_t(x_t - \hat{x}_t)$, for $t \geq 1$.

$$\sum_{t=1}^{T}\langle A_t, g_t \rangle = \frac{\beta_1}{1-\beta_1}\langle A_T, m_T \rangle + \sum_{t=1}^{T}\langle A_t, m_t \rangle + \frac{\beta_1}{1-\beta_1}\sum_{t=1}^{T-1}\langle A_t - A_{t+1}, m_t \rangle. \qquad (6)$$

After plugging in the value of $A_t$, (6) becomes

$$\sum_{t=1}^{T}\bar{\rho}\alpha_t\langle x_t - \hat{x}_t, g_t \rangle \leq \frac{\beta_1\bar{\rho}\alpha_T}{1-\beta_1}\langle x_T - \hat{x}_T, m_T \rangle + \sum_{t=1}^{T}\bar{\rho}\alpha_t\langle x_t - \hat{x}_t, m_t \rangle$$

$$+ \frac{\beta_1\bar{\rho}}{1-\beta_1}\sum_{t=1}^{T-1}\langle \alpha_t(x_t - \hat{x}_t) - \alpha_{t+1}(x_{t+1} - \hat{x}_{t+1}), m_t \rangle. \qquad (7)$$

LHS of this bound is suitable for applying Lemma 2 to obtain the stationarity measure. We have to estimate the three terms on the RHS. It is easy to bound the first term using Cauchy-Schwarz inequality and Lemma 1. Other two terms require longer estimations which we sketch below.

• Bound for $\frac{\beta_1\bar{\rho}}{1-\beta_1}\sum_{t=1}^{T-1}\langle \alpha_t(x_t - \hat{x}_t) - \alpha_{t+1}(x_{t+1} - \hat{x}_{t+1}), m_t \rangle$ in (7).

Decomposing this term gives

$$\langle \alpha_t(x_t - \hat{x}_t) - \alpha_{t+1}(x_{t+1} - \hat{x}_{t+1}), m_t \rangle = (\alpha_t - \alpha_{t+1})\langle x_{t+1} - \hat{x}_{t+1}, m_t \rangle + \alpha_t\langle x_t - x_{t+1}, m_t \rangle$$
$$+ \alpha_t\langle \hat{x}_{t+1} - \hat{x}_t, m_t \rangle.$$

For the first term, we use that $\alpha_t \geq \alpha_{t+1}$ and Cauchy-Schwarz inequality

$$\sum_{t=1}^{T-1}(\alpha_t - \alpha_{t+1})\langle x_{t+1} - \hat{x}_{t+1}, m_t \rangle \leq \alpha_1\hat{D}\sqrt{d}G.$$

For the second term we deduce by Cauchy-Schwarz inequality and nonexpansiveness of the projection

$$\alpha_t\langle x_t - x_{t+1}, m_t \rangle \leq \alpha_t^2\|m_t\|_{\hat{v}_t^{-1/2}}^2.$$

For the third term, we use Young's inequality to obtain the bound

$$\sum_{t=1}^{T-1}\frac{\beta_1\bar{\rho}}{1-\beta_1}\langle \alpha_t(x_t - \hat{x}_t) - \alpha_{t+1}(x_{t+1} - \hat{x}_{t+1}), m_t \rangle \leq \frac{\beta_1\bar{\rho}}{1-\beta_1}\alpha_1\hat{D}\sqrt{d}G + \sum_{t=1}^{T}\frac{\beta_1\bar{\rho}\alpha_t^2}{1-\beta_1}\|m_t\|_{\hat{v}_t^{-1/2}}^2$$

$$+ \sum_{t=1}^{T}\frac{\bar{\rho}-\hat{\rho}}{4}\|\hat{x}_{t+1} - \hat{x}_t\|_{\hat{v}_t^{1/2}}^2 + \frac{\bar{\rho}^2}{(\bar{\rho}-\hat{\rho})}\frac{\beta_1^2}{(1-\beta_1)^2}\sum_{t=1}^{T}\alpha_t^2\|m_t\|_{\hat{v}_t^{-1/2}}^2, \qquad (8)$$

• Bound for $\sum_{t=1}^{T}\bar{\rho}\alpha_t\langle x_t - \hat{x}_t, m_t \rangle$ in (7).

We proceed similar to [7], with a tighter estimation (resulting in the negative term on RHS) to obtain

$$\varphi_{1/\bar{\rho}}^{t+1}(x_{t+1}) \leq \varphi_{1/\bar{\rho}}^{t}(x_t) + \bar{\rho}\alpha_t\langle \hat{x}_t - x_t, m_t \rangle + \frac{\bar{\rho}}{2}\|\hat{x}_t - x_{t+1}\|_{\hat{v}_{t+1}^{1/2} - \hat{v}_t^{1/2}}^2 - \frac{\bar{\rho}-\hat{\rho}}{2}\|\hat{x}_t - \hat{x}_{t+1}\|_{\hat{v}_{t+1}^{1/2}}^2$$

$$+ \frac{\bar{\rho}}{2}\alpha_t^2\|m_t\|_{\hat{v}_t^{-1/2}}^2. \qquad (9)$$

Then we manipulate the fourth term on RHS with standard $\|a - b\|^2 \leq 2\|a\|^2 + 2\|b\|^2$, and Lemma 1,

$$\frac{\bar{\rho}}{2}\|\hat{x}_t - x_{t+1}\|_{\hat{v}_{t+1}^{1/2} - \hat{v}_t^{1/2}}^2 \leq \bar{\rho}\|\hat{x}_t - x_t\|_{\hat{v}_{t+1}^{1/2} - \hat{v}_t^{1/2}}^2 + \frac{G\bar{\rho}}{\sqrt{\delta}}\|x_t - x_{t+1}\|_{\hat{v}_t^{1/2}}^2$$

$$\leq \bar{\rho}\hat{D}^2\sum_{i=1}^{d}(\hat{v}_{t+1,i}^{1/2} - \hat{v}_{t,i}^{1/2}) + \frac{G\bar{\rho}}{\sqrt{\delta}}\alpha_t^2\|m_t\|_{\hat{v}_t^{-1/2}}^2. \qquad (10)$$

We use this estimation in (9) and sum to get

$$\bar{\rho}\alpha_t \sum_{t=1}^{T}\langle x_t - \hat{x}_t, m_t\rangle \leq \varphi_{1/\bar{\rho}}^1(x_1) - \varphi_{1/\bar{\rho}}^{T+1}(x_{T+1}) + \sum_{t=1}^{T}\left(\frac{1}{2} + \frac{G}{\sqrt{\delta}}\right)\bar{\rho}\alpha_t^2\|m_t\|_{\hat{v}_t^{-1/2}}^2$$

$$+ \bar{\rho}\hat{D}^2\sum_{i=1}^{d}\hat{v}_{T+1,i}^{1/2} - \sum_{t=1}^{T}\frac{\bar{\rho}-\hat{\rho}}{2}\|\hat{x}_t - \hat{x}_{t+1}\|_{\hat{v}_{t+1}^{1/2}}^2. \quad (11)$$

We collect (8) and (11) in (7). Finally, we have to obtain the stationarity criterion on the LHS of (7) by taking conditional expectation. This is not immediate due to coupling of $\hat{x}_t$, $\hat{v}_t$, and $g_t$. We use Lemma 2 to handle this issue and the negative term in (11) is utilized to cancel the third term in the RHS of the result of Lemma 2. Then, we use (5), plug in Lemma 4 and $\bar{\rho} = 2\hat{\rho}$ to conclude. $\qquad\square$

## 4 Applications & Extensions

### 4.1 Applications

**RMSprop**. The counterexamples presented in [28] show that RMSprop, similar to Adam might diverge in simple problems. Setting $\beta_1 = 0$ in AMSGrad [28] results in an algorithm similar to RMSprop, with the difference of having $\hat{v}_t$ as the output of the max step. Therefore, our analysis also applies to this version of RMSprop with similar guarantees.

**Corollary 1.** *Let $\beta_1 = 0$. Then, for a variant of RMSprop [28], Theorem 1 applies.*

It is easy to see that $\beta_1 = 0$ gives a better bound in Theorem 1. This is in fact common for the bounds of Adam-type algorithms even in the convex case [28]. Setting nonzero momentum parameters $\beta_1$, $\beta_2$ do not predict improvement, however, in practice they are routinely observed to improve performance.

**SGD with momentum**. When $\hat{v}_t = \mathbb{1}, \forall t$, AMSGrad reduces to an algorithm similar to SGD with momentum. Lack of diagonal step sizes in this case simplifies the analysis as weighted projections are not used in the algorithm. This specific case is studied in the recent work [26], with a slightly different way to set $m_t$. Our analysis can be seen as an alternative derivation of convergence for a method similar to [26].

**Constrained smooth optimization**. A special case of (1) is when $f$ is $L$-smooth. In this case, standard convergence measure is the gradient mapping [19], which is used in [5]: $\mathcal{G}_\lambda(x) = \hat{v}_t^{1/4}\lambda^{-1}(x - P_{\mathcal{X}}^{\hat{v}_t^{1/2}}(x - \lambda\hat{v}_t^{-1/2}\nabla f(x)))$. It is instructive to observe that when $\mathcal{X} = \mathbb{R}^d$, then $\|\mathcal{G}_\lambda(x)\| = \|\nabla f(x)\|_{\hat{v}_t^{-1/2}} \geq G^{-1}\|\nabla f(x)\|$ which is the stationarity measure for smooth unconstrained problems. When $\mathcal{X} \neq \mathbb{R}^d$, gradient mapping is used as the stationarity measure [7, 19, 26].

As illustrated in [7], for the specific case of constrained smooth minimization, norm of the Moreau envelope is of the same order as the norm of the gradient mapping, therefore, the results can be converted to guarantees on gradient mapping norms. Using similar ideas as in [12, Theorem 3.5], [7], one can show that $\|\mathcal{G}_{1/\bar{\rho}}(x_t)\| \leq C_{\text{g,m}}\|\nabla\varphi_{1/\bar{\rho}}^t(x_t)\|_{\hat{v}_t^{-1/2}}$, for a constant $C_{\text{g,m}}$ (see Appendix B.1).

### 4.2 Extension: Scalar AdaGrad with momentum

An alternative adaptive algorithm is AdaGrad [14] and its variants with first order moment estimation are referred to as AdamNC [28] or AdaFOM [4]. In unconstrained smooth stochastic optimization, it has been observed that the same proof structure applies to AMSGrad and AdaGrad-based methods simultaneously [4, 11]. However, in our setting, the analysis we developed for AMSGrad does not directly apply to AdaGrad-based methods.

The main reason is that $v_t$ in the case of AdaGrad does not admit a lower bound separated from $0$, unlike AMSGrad where $0 < \delta \leq \hat{v}_t$. The uniform lower bound is necessary for converting regular weak convexity assumption w.r.t. norm $\|\cdot\|$ to the one w.r.t. the weighted norm $\|\cdot\|_{v_t^{1/2}}$ in the sense of Remark 1. Naively assuming the existence of $\hat{\rho}$ in Remark 1 is not consistent, since $v_t$ is not separated from zero due to $v_t \geq \frac{\delta}{\sqrt{t}}$ in AdaGrad, and hence, the norm $\|\cdot\|_{v_t^{1/2}}$ is not well-defined.

In this section, we provide partial results on this direction. In particular, we show that the scalar version of AdaGrad, that is used in [22–24, 34], along with its variant with first order moment estimation also has the same convergence rate. In the framework of Algorithm 1, *scalar* (non-diagonal) version of these methods iterate as, for iid sampled $\xi_t \sim \mathsf{P}$, $E_{\xi_t}[g_t] \in \partial f(x_t)$,

$$\begin{cases} m_t = \beta_1 m_{t-1} + (1 - \beta_1) g_t \\ v_t = \frac{1}{t}(\delta + \frac{1}{d} \sum_{j=1}^t \|g_j\|^2) \\ x_{t+1} = P_{\mathcal{X}}(x_t - \frac{\alpha_t}{\sqrt{v_t}} m_t), \end{cases} \tag{12}$$

where $P_{\mathcal{X}}$ denotes standard Euclidean projection. The factor of $d^{-1}$ in front of gradient norms is to normalize the step size, as $\ell_2$-norm is dimension dependent. This factor only affects the dimension dependence of the bound. In this case, one does not need the time-dependent definitions for Moreau envelope and $\hat{x}_t$: $\hat{x}_t = \mathrm{prox}_{1/\bar{\rho}}(x_t)$ and $\varphi_{1/\bar{\rho}}(x) = \min_{y \in \mathcal{X}} f(y) + \frac{\bar{\rho}}{2}\|y - x\|^2$, due to lack of weighted projection. The proof then is similar to [7] with AdaGrad step sizes. The difficulties arising due to adaptive step sizes and existence of $\beta_1$, are handled using Lemmas 1, 3 and 4.

**Theorem 2.** *Let Assumption 1 hold. Then, for the method sketched in* (12)*, with* $\beta_1 < 1$*,* $\alpha_t = \frac{\alpha}{\sqrt{t}}$

$$\mathbb{E}\|\nabla\varphi_{1/2\rho}(x_{t^\star})\|^2 \leq \frac{2G\sqrt{\rho}}{\alpha\sqrt{T}}\left[C_1 + \left(1 + \log\left(\frac{TG^2}{\delta} + 1\right)\right)C_2\right],$$

*where* $C_1 = \varphi_{1/2\rho}(x_1) - f^\star + \frac{12\alpha d G^2}{\sqrt{\delta}(1-\beta_1)}$*,* $C_2 = \frac{8\rho\alpha^2 d}{(1-\beta_1)^2}$*.*

The terms in the bound are simplified as Theorem 1 and their non-simplified variants are in the proof in Appendix A. Our analysis takes care of the coupling between the AdaGrad step sizes and iterates of the algorithm. However, unlike our results with AMSGrad, in this case the analysis does not cover the diagonal case, where the step sizes are set by using elementwise squared gradients $g_t^2$. We leave it as an open question to derive similar results for AdaGrad-based methods with diagonal step sizes.

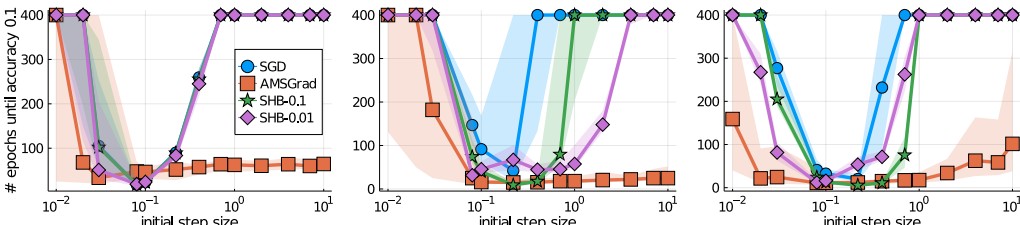

Figure 1: left-right: $\kappa = \{1, 10, 100\}$. Number of epochs to $f(x) - f^\star \leq 0.1$ vs. initial step size

## 5 Numerical experiment

This section illustrates the potential advantages of adaptive algorithms, in particular AMSGrad, for solving a prototypical weakly convex problem, compared to SGD and SGD with momentum (also referred in [26] as stochastic heavy ball: SHB). As popular in the literature of weakly convex stochastic optimization methods [8, 10, 26], we will compare the algorithms in terms of their robustness to initial step sizes. We note that *"robustness to tuning"* of algorithms is also investigated in the context of deep learning in the literature and the advantage of adaptive algorithms such as Adam/AMSGrad is observed [6, 31]. In particular, we solve the robust phase retrieval problem [9, 16, 17]: $\min_{x \in \mathbb{R}^d} \frac{1}{n}\sum_{i=1}^n |\langle a_i, x\rangle^2 - b_i|$, where $A = [a_1, \ldots, a_n]^\top \in \mathbb{R}^{n \times d}$, $n = 300$, $d = 50$. Weak convexity of this problem is well-known [7, 9].

We briefly recall the setup from [26] that considered SGD with momentum for solving this problem. The data is generated as $A = QD$, with a standard normal distributed matrix $Q \in \mathbb{R}^{n \times d}$ and $D = \mathrm{linspace}(1/\kappa, 1, d)$, where $\kappa \geq 1$ controls the conditioning. We generate the solution $x^\star$ as a standard normal random vector with unit norm. The observations are generated as $b = Ax^\star + \delta\eta$

---

[2]We make the "effective initial step sizes" of algorithms equal. In particular we pick $\alpha_0^{\mathrm{SGD}} = \alpha_0^{\mathrm{MSGD}} = \frac{\alpha_0^{\mathrm{AMS}}}{\beta_2\sqrt{\max_i(g_{1,i}^2)}}$, since the initial step size of AMSGrad is $\frac{\alpha_0}{\sqrt{\hat{v}_1^2}}$.

where elements of $\eta \in \mathbb{R}^n$ have distribution $\mathcal{N}(0, 25)$ and $\delta = \text{diag}(\delta_1, \ldots, \delta_d)$ is such that $|\{i \in [n] : \delta_i = 1\}|/n = 0.2$, which means that $20\%$ of the observations are corrupted.

With this setup, it is proven in [9, Lemma B.12] that only solutions of the problem are $\{x^\star, -x^\star\}$. Therefore, for the algorithms, we will use $f(x_k) - f(x^\star) \leq \varepsilon$ as the stopping criterion. We run stochastic subgradient method (SGD) [7], momentum SGD (SHB) [26] and AMSGrad that we analyzed in this paper. For all algorithms, the step size is chosen as $\alpha_k = \alpha_0/\sqrt{k}$. We varied the initial step size[2] between $0.01$ and $10$ for all algorithms, and we plotted the number of epochs that the algorithms take to reach to $f(x) - f(x^\star) \leq 0.1$. In terms of other algorithmic parameters, we use both $\beta = 0.1$ and $\beta = 0.01$ for SHB, as in in [26] and $\beta_1 = \beta_2 = 0.99$ as popular, for AMSGrad.

We present the results in Figure 1 for varying values of $\kappa = \{1, 10, 100\}$, where each setup is run for $50$ times, medians are drawn as lines and the region between 20th and 80th percentiles are shaded. It has been observed in [26] that SHB improves the robustness of SGD to initial step sizes. We observe in Figure 1 that AMSGrad shows a more robust behavior to initial step size compared to both algorithms. We note that our findings indicate the potential of AMSGrad and adaptive methods for weakly convex optimization. Moreover, our findings about robustness of adaptive algorithms to tuning is consistent with the findings from deep learning literature [6, 31].

## Acknowledgments and Disclosure of Funding

Most of the work was done while Ahmet Alacaoglu and Yura Malitsky were at EPFL.

This project received funding from NSF Award 2023239; DOE ASCR under Subcontract 8F-30039 from Argonne National Laboratory; the Wallenberg Al, Autonomous Systems and Software Program (WASP) funded by the Knut and Alice Wallenberg Foundation, with the project number 305286; the European Research Council (ERC) under the European Union's Horizon 2020 research and innovation programme (grant agreement no 725594 - timedata); the Swiss National Science Foundation (SNSF) under grant number 200021_178865/1; the Department of the Navy, Office of Naval Research (ONR) under a grant number N62909-17-1-2111; and the Hasler Foundation Program: Cyber Human Systems (project number 16066).

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
