# OpenReview forum: "Convergence of adaptive algorithms for constrained weakly convex optimization"
_NeurIPS.cc/2021/Conference — NeurIPS 2021 Poster_

### Official Review · Reviewer_SLqj · 2021-07-15

**Rating:** 7
**Confidence:** 4

**Summary:**

This paper presents an adaptive gradient descent method for a convex set-constrained weakly convex (hence, nonconvex and possibly nonsmooth) optimization problem.

**Limitations And Societal Impact:**

Boundedness assumption on the stochastic gradient, even though standard in adaptive optimization, is quite strong for unbounded domain set. Many works get around this difficulty by assuming compactness of set \mathcal{X}. Since the paper also briefly discussed bounded domains, I believe that this is not a major issue.
Overall, I think the paper makes a good contribution. Given that this is the first paper looking into an adaptive procedure for nonsmooth nonconvex problems, the authors could have done a better job in highlighting the key aspect of the analysis which allowed for providing convergence.


**Main Review:**

The authors used AMSGrad proposed by Reddi et al. [2018] for showing their results. AMSGrad was a modification of ADAM that exhibited provable convergence. The authors also used Moreue envelop-based analysis in Davis et al. [2018] where the norm is suitably weighted for the adaptive procedure. The bounds provided in Theorem 1 are very similar to state-of-the-art results for adaptive algorithms like Adam, AMSGrad, RMSProp, etc.
The Paper was quite easy to read and the authors did a good job in providing key insights into the proof techniques.

**Time Spent Reviewing:**

6 hours

---

> ### Author Response · Authors · 2021-08-10
> **Response to Reviewer SLqj**
>
> We are grateful to the reviewer for the careful reading of our manuscript and thoughtful comments.
>
> > *"the authors could have done a better job in highlighting the key aspect of the analysis which allowed for providing convergence."*
>
> We will highlight better the key aspect of the analysis which allowed for proving convergence.

---

### Official Review · Reviewer_NSxc · 2021-07-16

**Rating:** 7
**Confidence:** 4

**Summary:**

This paper analyzes AMSGrad when applied to a broad family of nonsmooth, nonconvex, stochastic, constrained optimization problems. Convergence rates based on the norm of the Moreau envelope's gradient (although in a rescaled norm) are presented. The rates nearly match the O(1/T^{1/4}) rate of the subgradient method, off by a logarithmic term. The primary innovation in this work is tackling technical barriers to handle first and second order moment parameters that complicate analysis.

**Limitations And Societal Impact:**

The convergence guarantee's given in the rescaled norm based on $v_{t}$ and based on the rescaled Moreau/proximal operator using $v_t$. As a result (and as the paper acknowledges in remark 2), extra care is needed to understand the results in the given norm $\|\cdot\|$. Particularly, the best guarantee given from the proposed analysis is worse by a non-negligible factor of $1/\delta$. Establishing some lower bounding theory for $v_t$ may overcome this current limitation.

**Main Review:**

-The terminology used for describing convergence rates is somewhat inconsistent. The abstract uses the more modest description of a O(1/T^{1/4}) rate in the gradient norm whereas the main text consistently frames itself as a O(1/\sqrt{T}) rate in terms of the near stationarity condition (5), which hides a factor of two.

-Algorithm 1 currently does not define $\xi_t$ which is presumably sampled i.i.d. at each iteration.

-The analysis implicitly uses a sum rule for nonconvex nonsmooth calculus that requires additional regularity assumptions to be assumed. Namely, it need not be the case that selecting $g_t\in \partial f(x_t,\xi_t)$ for each realization of $\xi_t$ produces an unbiased estimator of a element $q_t$ in $\partial f(x_t)$ (as is claimed in Lemma 2). Regularity conditions on each $f(x,\xi)$ are needed. It would suffice to additionally assume that each $f(x,\xi)$ is locally Lipschitz everywhere on an open set containing $X$.

-Remark 2 currently claims $\|x_t-\hat x_t\|^2_{v_t^{1/2}} \geq \sqrt{\delta}\|x_t-\hat x_t\|^2$. I believe the $\sqrt{\delta}$ should be $\delta$.

-Relaxing the $\|\cdot\|_\infty$ norm subgradient constraint to instead be a bound on the expected norm seems more difficult than the work suggests. Doing so would remove the uniform upper bound on the size of $v_t$, which would then make converting guarantees back to the original norm $\|\cdot\|$ much harder.

**Time Spent Reviewing:**

5

---

> ### Author Response · Authors · 2021-08-10
> **Response to Reviewer NSxc**
>
> We are grateful to the reviewer for the careful reading of our manuscript and thoughtful comments.
>
> > *"The terminology used for describing convergence rates is somewhat inconsistent."*
>
> We will adapt the terminology accordingly.
>
> > *"Algorithm 1 currently does not define $\xi_t$"*
>
> Indeed, $\xi_t$ is sampled iid at each iteration, we will add the explanation.
>
> > *"The analysis implicitly uses a sum rule for nonconvex nonsmooth calculus that requires additional regularity assumptions to be assumed."*
>
> We are thankful for the reviewer for pointing out the impreciseness. We will change the assumption to: picking $g_t \in \partial f(x_t, \xi_t)$ such that $\mathbb{E} [g_t] \in \partial f(x_t)$.
>
> > *"In Remark 2, I believe the $\sqrt{\delta}$ should be $\delta$."*
>
> We believe that $\delta$ dependence is correct as $v_{t, i} \geq \delta$ and the norm we have in Remark 2 is $\| \cdot \|_{v_t^{1/2}}^2$.
>
> > *"Relaxing the assumption on subgradient norm to be a bound on the expected norm seems more difficult than the work suggests."*
>
> We agree with the reviewer that one should be careful when switching assumptions. We believe that we can use Hölder’s inequality similarly as in [36, Lemma 27] to achieve the result with the slightly worse rate.
>
> > *"Establishing some lower bounding theory for $v_t$ may overcome this current limitation."*
>
> We would like to mention that such dependence in $1/\delta$ is common in nonconvex analyses in the literature for adaptive algorithms (please also see lines 176-179). We totally agree with the reviewer that overcoming this limitation is important.

---

> > ### Comment · Reviewer_NSxc · 2021-08-18
> > **Reply to Authors**
> >
> > Thank you for your careful reply addressing the small mathematical rigor concerns raised. My score (of Good paper, accept) remains unchanged.

---

### Official Review · Reviewer_kQU3 · 2021-07-21

**Rating:** 6
**Confidence:** 4

**Summary:**

The paper provides a convergence analysis for the AMSGrad algorithm for the more general class of functions that are weakly convex. The convergence guarantee is comparable to existing results for SGD, and the main advantage is that the AMSGrad algorithm does not require knowledge of the weak convexity parameter and it automatically adapts to it. The paper extends the convergence analysis to other algorithms, such as SGD with momentum and the AdaGrad algorithm with scalar step sizes.


**Limitations And Societal Impact:**

Yes

**Main Review:**

Significance:
Adaptive methods in the AdaGrad/Adam family are widely used in practice, in both the convex and non-convex setting. Prior analyses for these algorithms in the non-convex setting assume smoothness. This paper extends this line of work by considering weakly convex functions, which are more general than smooth functions. In addition to adapting to the weak convexity parameter, another strength of the analysis is that it works for constant beta_1 and beta_2 parameters, which is in line with how Adam is used in practice.

One of the weaknesses of the analysis is that it strongly relies on the value delta used to initialize the preconditioner, which is crucial for bounding key error terms, resulting in a dependence of 1/sqrt(delta) in the convergence. This limits the applicability of the analysis approach and it does not allow for arbitrarily small initializations as used in practice.

The computational evaluation shows the benefits of using AMSGrad compared to using non-adaptive methods such as SGD. However, the experiments consider small synthetic instances and thus the experimental evaluation is limited. Overall, this is a good contribution to this line of work.

Novelty/originality: The paper builds on prior analyses of SGD for weakly convex functions as well as an an analysis of AMSGrad for convex functions that works for constant beta_1 and beta_2 parameters. The non-convexity together with the adaptive and stochastic step sizes poses some additional challenges in the analysis compared to prior work, and the main novelty in this work is to bound the new error terms that arise. Overall, there is sufficient novelty but it is perhaps somewhat limited.

Clarity: The paper is reasonably clear.

**Time Spent Reviewing:**

I did not track the hours

---

> ### Author Response · Authors · 2021-08-10
> **Response to Reviewer kQU3**
>
> We are grateful to the reviewer for the careful reading of our manuscript and thoughtful comments.
>
> > *"One of the weaknesses of the analysis is that it strongly relies on the value delta used to initialize the preconditioner, which is crucial for bounding key error terms, resulting in a dependence of 1/sqrt(delta) in the convergence. This limits the applicability of the analysis approach and it does not allow for arbitrarily small initializations as used in practice."*
>
> We agree with the reviewer that the value of $\delta$ plays an important role in our analysis and our bounds have polynomial dependence on $1/\delta$. On the other hand, we wish to highlight that dependence on $1/\delta$ is common in the existing analyses of Adam-type methods for solving nonconvex problems (as we mentioned in lines 176-179). Except one reference ([11]), the other works have a polynomial dependence on $1/\delta$. We believe it is interesting to investigate how to improve this dependence for nonconvex analyses. As we explained in lines 176-179, the technique of [11] does not seem to apply in our case to improve the dependence.
>
> > *"However, the experiments consider small synthetic instances and thus the experimental evaluation is limited."*
>
> Indeed, our numerical experiments are merely to showcase the promise of adaptive methods for weakly convex problems. As we believe that the merit of adaptive methods for neural network training is well-known, we do not include numerics in this setting.

---

### Decision · Program_Chairs · 2021-09-27

**Decision:**

Accept (Poster)

**Comment:**

The paper introduces an adaptive method for weakly convex optimization, which generalizes the setting of smooth nonconvex optimization. A particularly important aspect of the introduced method is that it automatically adapts to the weak convexity parameter, without any prior knowledge. The paper makes a good contribution to the literature on adaptive gradient methods (similar to the very popular AdaGrad) and all the reviews expressed support in seeing this paper published at NeurIPS.